# Accurate and reproducible enumeration of CD4 T cell counts and Hemoglobin levels using a point of care system: Comparison with conventional laboratory based testing systems in a clinical reference laboratory in Cameroon

Bertrand Sagnia[1]*, Fabrice Mbakop Ghomsi[2], Sylvie Moudourou[3], Ana Gutierez[4], Jules Tchadji[1], Samuel Martin Sosso[3], Alexis Ndjolo[1], Vittorio Colizzi[5]

1 Laboratory of Microbiology and Immunology of CIRCB, Yaounde, Cameroon, 2 Centre de Santé Catholique de Nkolondom, Yaounde, Cameroon, 3 Medical Analysis Laboratory of CIRCB, Yaounde, Cameroon, 4 Catholic Hospital of BIKOP, Mbalmayo, Cameroon, 5 Laboratory of Immunology, Faculty of Sciences, University of Rome "Tor Vergata", Rome, Italy

* bertrandsagnia@yahoo.fr

## Abstract

### Background

Measurements of CD4 T cells and hemoglobin (Hb) are conventionally used to determine the immunological state and disease progression for HIV-infected patients. We obtained a small lightweight point-of-care device, the BD FACSPresto™ in order to demonstrate its ability to deliver CD4 and Hb analysis in comparison with two larger clinical machines the BDFACSCanto™ analyzer and Sysmex XN 1000 haematology analyzer. The advantages of using the POC device include access to HIV patient data in remote and in resource limited settings.

### Method

The analytical performance of the BD FACSPresto™, compared with the FACSCanto™ II flow cytometer and the Sysmex XN 1000 haematology analyzer was evaluated by testing 241 routine clinical specimens collected in EDTA tubes from patients attending the Immunology and Microbiology laboratory of Chantal BIYA International Reference Centre (Yaounde, Cameroon) between January and May 2016.

### Results

The mean in absolute counts and percentage of CD4 T cells was 606 cells/mL and 25% respectively via the FACSPresto™, and 574 cells/mL and 24% respectively via the BD FACSCanto™ II. The mean concentration of Hb levels was 11.90 on the Sysmex XN 1000 and 11.45 via the BD FACSPresto™, A high correlation ($R^2 = 0.95$, $P < 0.001$) of Hb level

**Data Availability Statement:** All relevant data are within the manuscript and its Supporting Information files.

**Funding:** The author(s) received no specific funding for this work.

**Competing interests:** The authors have declared that no competing interests exist.

**Abbreviations:** ART, Antiretroviral therapy; BD, Becton Dickinson; CIRCB, Chantal BIYA International Reference Centre for Research on HIV/AIDS; EDTA, Ethylene diamine tetraacetic acid; Hb, Hemoglobin; HIV, Human Immunodeficiency Virus; ISO 15189–2012, International Organization for Standardization of medical laboratories; MOH, Ministry of Health; NIH, National Institute of Health; POC, Point Of Care; QASI, Quality Assessment and Standardization of Indicators); TAT, Turn Around Time; TBNK, measurements: T lymphocytes, B lymphocytes and Natural Killer lymphocyte measurements; UKNEQAS, United Kingdom National External Quality Assurance service for CD4 testing; WHO, World Health Organization.

measurements was noted between the BD FACSPresto™ and Sysmex XN 1000 hematology analyzer. Overall, a Bland-Altman plot of the differences between the two methods showed an excellent agreement for absolute and percentage CD4 counts and hemoglobin measurements between POC and conventional methods evaluated here. Furthermore, the study demonstrated the ease of use of the BD FACSPresto™ POC technology in remote areas.

## Conclusion

The BD FACPresto™ is a suitable tool for CD4 enumeration in resource-limited settings, specifically providing a deployable, reliable POC testing option. The BD FACSPresto™ performed appropriately in comparison to the conventional reference standard technologies. The BD FACSPresto™, system provides accurate, reliable, precise CD4/%CD4/Hb results on venous blood sampling. The data showed good agreement between the BD FACSPresto™, BD FACSCanto™ II and Sysmex XN 1000 XN 1000 systems.

## Introduction

In the context of HIV, CD4 T cells are infected with the virus and their number decreases [1]. Globally in 2018, more than 38 million people died because of HIV infection and the key cells infected [2] are CD4 T cell lymphocytes. Phenotyping of lymphocytes using flow cytometry is extensively used [3, 4] for enumeration of the CD4 absolute cell counts to determine HIV-infection or AIDS disease status, for monitoring disease progression or co-infection, for patient staging, and for initiation of antiretroviral treatment (ART).

Anemia has been identified as an additional parameter to assess HIV-disease progression and can be diagnosed by measuring the concentration of hemoglobin (Hb) in venous blood [5]. Nutrient deficiencies, exposure to antiretroviral drugs such as zidovudine or concomitant conditions could be associated with anemia. Close monitoring of pregnant women with preexisting anemia and advanced HIV/AIDS is recommended [6, 7].

The burden of HIV infection is significant in countries where infrastructure is lacking or absent. In this context, many pharmaceutical companies proposed new instruments with innovative characteristics such as point of care (POC) utility able to enumerate only CD4 absolute cell counts like the Abbott PIMA (previously Alere) or CD4 absolute counts and percentages combined with hemoglobin concentration like the FACSPresto™, from Becton Dickinson [8–10]. Many performance comparison studies have been done with the BD FACSCalibur™ as a validated reference flow cytometry instrument in different countries in Africa [11–14] or in Asia [15]. To date, there have been no studies published using the flow cytometer BD FACSCanto™ II, with clinical software to measure the T/B/NK cell populations and other parameters for research and clinical analysis. It is hypothesized that results for CD4% and Hb on the BD FACSPresto™, system are accurate and reproducible when samples are measured within 24 hours of phlebotomy.

The objective of the study was to demonstrate that the performance of the BD FACSPresto™ system is comparable to that of a Sysmex XN 100 hematology analyzer and to BD FACSCanto™ II flow cytometry with clinical software. Both systems are used respectively, to determine hemoglobin concentration and absolute cell counts and percentages of CD4 T

lymphocytes stained with BD Multitest Reagents in BD Trucount tubes using venous blood samples from patients.

## Materials and methods

This study was conducted at the Chantal BIYA International Reference Centre for research on prevention and management of HIV/AIDS (CIRCB), the CD4 reference laboratory in Yaounde, Cameroon according to flow chart in Fig 1. The venous blood specimens were compared using conventional laboratory technologies BD FACSCanto[TM] II and Sysmex XN 1000 hematology analyzers and by using the POC device, BD FACSPresto[TM],. Blood samples were obtained from patients undergoing routine CD4 monitoring at the reference laboratory. All specimens were processed within 24 hours of collection. EDTA venous blood samples were analyzed on the BD FACPresto[TM], BD FACSCanto[TM] II and Sysmex XN 1000 hematology analyzer. Results obtained from the BD FACSPresto[TM], device were intended for research use only and were not used for clinical patient management. All procedures were conducted under good clinical laboratory practices and good clinical practice guidelines to ensure quality of laboratory testing, safety and confidentiality of subject's participation in the study.

### Samples

Venous blood (4mL) was collected in K3-EDTA tubes and gently inverted several times to ensure proper mixing. Samples were excluded from analysis if there were instrument problems or failure to process as per protocol. Exclusion criteria included the following: poor sample quality, process controls were not tested prior to sample testing, lymphosum failure, samples processed outside the time window for sample staining and/or acquisition, acquisition did not satisfy the minimum number required of lymph events, beads, and time and reagent storage issues.

### Analysis on BD FACSPresto[TM]

Venous blood specimens collected were analyzed on the BD FACSPresto[TM], machine. For analysis on the FACSPresto[TM], a drop of blood from a Pasteur pipette was loaded onto the FACSPresto[TM], cartridge capped and incubated at room temperature for 18 minutes; following incubation the cartridge was loaded onto the analyzer. On each day, prior to testing, the BD FACSPresto[TM], instrument was turned on, the instrument quality control (QC) test automatically run, and results printed. The CD4 and Hb external quality controls were run on the corresponding instruments before testing patient samples. Specimens with valid results were analyzed. Results were considered invalid if testing did not comply with the protocol procedures (inclusion or exclusion criteria, testing outside the recommended time window or if system errors suppressed results). The Lab Technicians and Nurses from the medical section received one day training before the evaluation by the manufacturer.

### Analysis on BD FACSCanto[TM] II

This BD flow cytometer is an in vitro diagnostic instrument, housed at the Reference laboratory of CIRCB. The analysis on the FACSCanto[TM] II was performed on a machine with clinical software for CD4 count. Fluorescent beads for instrument QC (BD CS&T beads and BD FACS 7-color Setup Beads) were used. In brief, 20μl of BD Multitest fluorescent conjugated monoclonal antibodies, and 50μl of whole blood were added to the TruCOUNT tube and vortexed for 5 seconds. The Multitest consists of CD3-FITC/CD8-PE/ CD45-PerCP/CD4 APC reagent. The mixture was incubated for 15 minutes at room temperature in the dark before adding

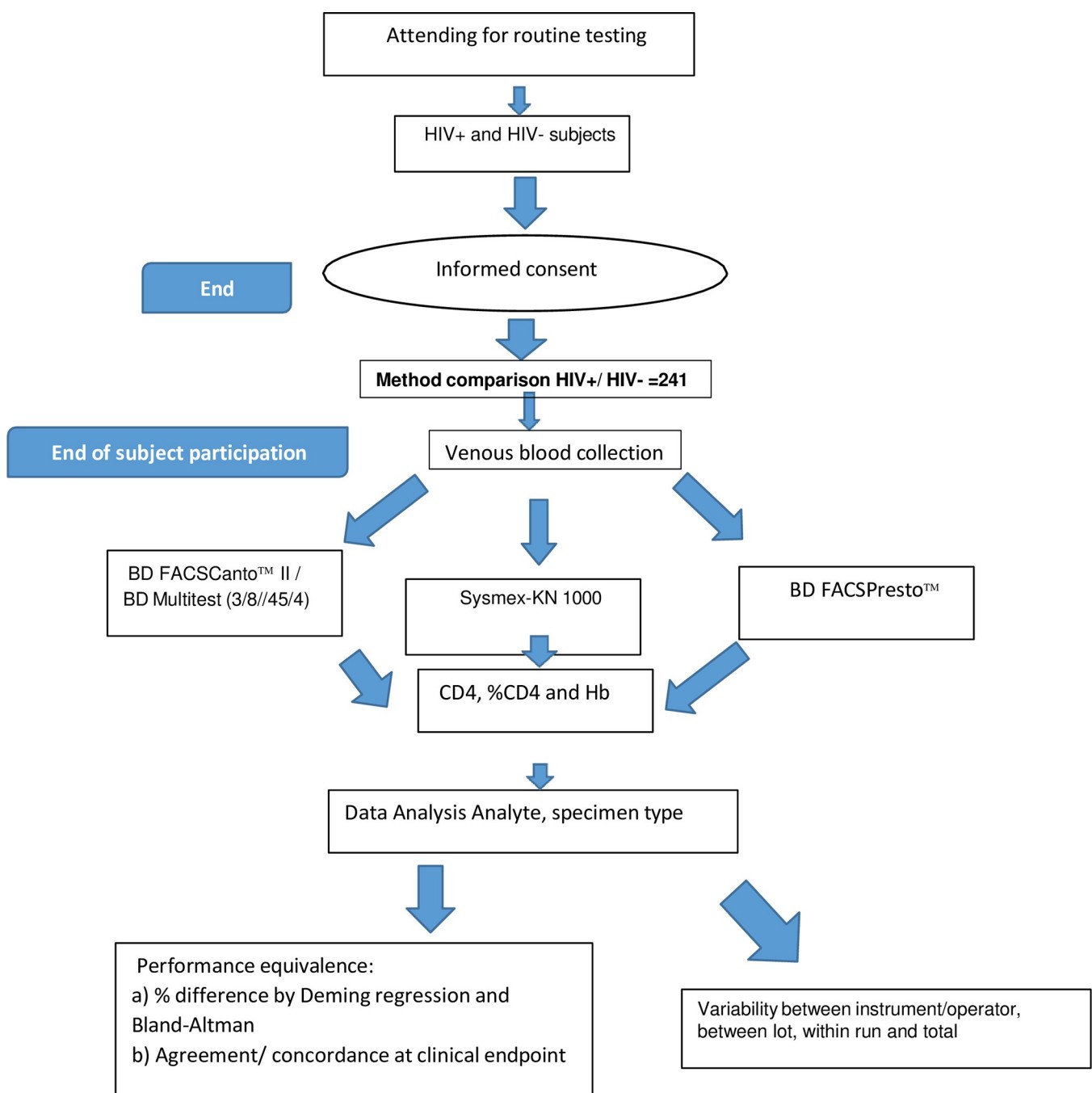

**Fig 1. Clinical evaluation of the BD FACSPrestoTM system flowchart.** Evaluation of the performance of the BD FACSPrestoTM system using venous blood specimens from subjects attending a routine clinic visit.

450μl of FACS™ lysing solution containing 15% formaldehyde and 50% diethylene glycol (BD Biosciences, San Jose, CA) and incubating for an additional 15 minutes in the dark prior to acquisition on the BDFACSCanto$^{TM}$ II using Clinical Software with automated gating and analysis. The operator had received appropriate three to five days training on the reverse pipetting technique and on the performance of the assay by manufacturer's guidelines prior to initiation of the evaluation. Internal quality control was monitored routinely. Laboratory staff

participate in an external quality assurance program with the Cameroon National Quality Assurance program for CD4 enumeration in collaboration with QASI® (Quality Assessment and Standardization of Indicators) in Canada and the UKNEQAS (United Kingdom National External Quality Assurance service) for CD4 testing.

### The Sysmex XN 1000 hematology analyzer

To determine the percentage of hemoglobin, the Sysmex XN 1000 automated Hematology analyzer was used according to manufacturer instructions. The concentration of hemoglobin was measured directly from the venous blood tube. Three levels of the Hb controls (low, normal, and high levels) were used. Sysmex control tests were used for quality control assurance.

### Statistical methods

CD4+ T cell counts obtained from the FACSPresto™ device were compared to counts collected on the FACSCanto™ II. The amount of Hb obtained from FACSPresto™ device was compared to levels measured on the Sysmex XN 1000 Analyzer at CIRCB. Descriptive statistics were used for the data. Differences in parameters between the two groups were determined by Wilcoxon signed rank test and paired t-test. Passing-Bablok regression was used for the method correlation and correlation coefficients were determined. To determine the bias between the platforms, Bland-Altman analysis was done. The bias was defined as the mean difference between two methods. The limits of agreement were calculated as the mean±1.96 Standard Deviation (SD) of the differences of the results obtained. Confidence intervals for the bias and the limits of agreement were calculated. Pollock analysis was done to calculate the relative bias and the limits of agreement, which were defined as the mean±1.96SD of the relative mean bias of paired measurements. The data was plotted with the y-axis representing the percentage difference relative to the absolute value (x-axis) of the comparator test. The percentage similarity between a sample pair was determined and defined as the average between the methods divided by the comparator method multiplied by 100. The analysis was done for comparing CD4% values on the FACSPresto™ and the BD FACSCanto™ II.

### Ethical review

The Cameroon National Ethics Committee approved the protocol prior to implementation with the number 2016/08/754/L/CNERH/SP. Written informed consent was required for residual samples from routine CD4 testing services. Residual blood from routine testing was used for the analysis. No personally identifiable information was made available to the researchers.

## Results

A total of two hundred and forty one participants were enrolled in this technical accuracy enumeration. Approximately 75% of enrolled participants were female, while 60% had a CD4+ T cell count above 350 cells/μL, as measured by the BD FACSCanto™ II. The mean and median ages of enrolled participants were 36 years with 53% of participants aged between the ages of 25 and 65 years. Almost 25% of the BD FACSPresto™ tests were sampled and run by nurses for reproducibility. The temperature range of the health care facility where POC tests were run was between 22–27˚C with a mean and median of 24˚C. Samples tested using the BD FACSCanto™ II had a median of 447 cells/μL (range: 11–4,140cells/ul), compared with a median of 468 cells/ul (range: 4–5,098 cells/ul) on the BD FACSPresto™, p <0.02. About 16% (n = 39) of all participants were HIV negative according to the Cameroon national HIV testing algorithm. HIV negative participants were mostly female, (n = 30) with an overall median CD4

**Table 1. Characteristics of study participants.**

|  | Male | Female | Total |
|---|---|---|---|
| Number of subjects (%) | 61 (25) | 180 (75) | 241 (100) |
| Median age | 36 | 36 | 36 |
| Median CD4 cells/µL (BD FACSCanto[TM]II) | 346 | 605 | 447 |
| CD4< 100 cells/µL (%) | 8 (13,11) | 13 (6,70) | 21 (8,24) |
| 100<CD4<350 cells/µL (%) | 22 (36,07) | 50 (25,77) | 72 (28,24) |
| 350<CD4<500 cells/µL (%) | 11 (18,03) | 34 (17,53) | 45 (17,65) |
| CD4>500 cells/µL (%) | 20 (32,79) | 90 (46,39) | 110 (43,14) |

count up to 500 cells/µL. The participant distributions and CD4+ T cell thresholds are found in Table 1.Valid results were observed for 233 specimens for BD FACSPresto[TM] and 230 for BD FACSCanto[TM] II. Eleven samples were excluded from analysis because of instrument problems or non-respect of protocol as described above. The FACSCanto II and FACSPresto classified respectively 39% and 36% of enrolled patients above 350 cells/µl. Before the test and treat initiative, quantitative CD4 testing was used to assess patients' eligibility for reflex crypto-coccal testing or ART eligibility depending on national guidelines. The BD FACSPresto[TM] had a sensitivity of 85.36% (83.3–91.1%) and specificity of 96.90% (93.1–97.9%) compared to the BD FACSCanto[TM] II to correctly identify patients eligible for ART at the threshold of 350 cells/µL.

The comparison of hemoglobin values between the BD FACSPresto[TM] and Sysmex 100 was evaluated and the percentage of similarity on 119 samples was more than 99% Table 2. The results for the two samples of CD4 (low and medium) for external quality control with QASI from the Canadian Public Health Agency were as follow: 155 cells/mL and 357 cells/mL for FACSCantoII[TM] and 111 cells/mL and 357 cells/mL for FACSPresto[TM].

The Deming regression results for the CD4 absolute count, CD4 percentage and the rate of hemoglobin gave $R^2 = 0.98$, $R^2 = 0.94$ and $R^2 = 0.96$ respectively (Fig 2). Fig 3 illustrates the strong linear correlation between CD4 enumeration machines and hemoglobin instruments.

Performance comparisons were done between the BD FACSPresto[TM] system and Sysmex XN 1000 automatic hematology analyzer for the measurement of Hb concentrations. For Hb concentrations in venous blood samples, we observed significant correlations p<0.001 between the values generated by the BD FACSPresto[TM] system and the reference standard, where slope values were 0.91 and R2 value of 0.98, for all participants. In addition, approximately≥95.8% of participants were within the mean±1.96 SD of the relative deviation.

## Discussion and conclusions

The CIRCB is a large reference laboratory working under the International Organization for Standardization of Medical Labs (ISO 15189–2012). Before using new equipment and its

**Table 2. FACSPresto[TM] comparing with Sysmex XN 1000 on the Haemoglobin concentration.**

|  | BD FACSPresto[TM] venous vs Sysmex XN 1000 XT (venous) |
|---|---|
| N | 119 |
| Hb dl/µL BD FACSPresto[TM] (range) | 12 (6,5–16) |
| Hd dl/µL Sysmex XN 100 (range) | 12 (6,6–16,1) |
| Coefficient of determination $R^2$ | 0,94 |
| Absolute mean bias (dl/µL) (LOA) | -0,419 (-0,578, -0,261) |
| Percentage of similarity | 99,1% |

implementation throughout the country, a comparison of results generated by a new instrument with the existing gold standard must be performed. Before the "test and treat" initiative, more than 10 million of patients infected with HIV needed of feasible decentralized services like early infant diagnosis (EID), CD4 T cells count measurement, viral load and genotyping testing with POC testing (POCT) for life saving. The CD4+ T-lymphocyte count remain relevant as one of the biomarkers to identify patients who are highly immunocompromised for optimal management and monitoring of HIV-infected patients on ART especially in resource limited settings. The BD FACSPresto™ is a POC instrument capable of doing CD4 absolute and percentage counts as well as hemoglobin levels. The BD FACSCanto™ II is the existing instrument at CIRCB for T/B/NK measurements [16] for CD4 enumeration of HIV infected patients throughout Cameroon. Installing a POC instrument in remote areas with consistent performance characteristics is very important. According to WHO and NIH issued guidelines recommending initiation of ART regardless of the CD4 cell counts and recommending use of viral load for monitoring HIV subjects [17], CD4 enumeration is still important in the management of HIV infected patient in rural areas of developing countries where not all patients have access to viral load testing. In addition, there is low adoption of viral load as a routine central laboratory test for patients-staging where ART availability is limited [18]; therefore, enumeration of the CD4 cells counts still has a role when viral load is not easily available. To add, CD4 results contribute to the interpretation of HIV genotyping testing to maintain or to change the ongoing ARV regimen. The CD4 level can often indicate the urgency of putting the patient on a treatment with strong antiviral activity sometimes quadruple therapy) in order to quickly reduce viremia and restore the CD4 level [19]. Furthermore, HIV-infected individuals have higher risks of co-morbidity and are more vulnerable to opportunistic infections [20]. The present study is the first published paper to demonstrate that BD FACSPresto™ provides excellent results when compared with a BD FACSCanto™ II clinical machine, showing very good agreement between both instruments. This study supports that CD4 T cells measurements were not influenced by the counting method used. The BD FACSPresto™ results were reproducible when operated by trained operators, such as nurses after minimum training. It is easy to use and attest to the fact that BD FACSPresto™ can accurately identify those patients with ART failure irrespective of technical laboratory expertise.

The BD FACSPresto™ analyzer has the advantage of having the complete incubation process outside the machine and it can test approximately 60–80 samples daily. The analyzer is an easy-to use instrument, providing daily internal quality controls before starting the assay. The analyzer provides absolute, percentage of CD4+ cell counts as well as hemoglobin levels [15] which are all useful for monitoring the pediatric population and tracking anemia in pregnancy. This instrument has a battery and can run without main power for about six hours. It is easy to carry in rural settings and easy to use also for personnel not working in the lab. Placing this instrument in rural areas will reduce the loss to follow-up of patients and will support management of their infections. This POC technology is ideal for cross-training lab staff and represents a breakthrough solution to the most basic operational challenges of flow cytometry guaranteeing the independence of non-expert staff.

Given the introduction of POCs in rural areas for viral load and EID, their validation could be a valuable tool for laboratories undergoing the accreditation process [21, 22]. The BD FACSPresto™ could improve the standardization of results which remains a critical point to make clinical diagnoses comparable between different laboratories. Based on internal experience, FACSPresto™ and FACSCanto™ II differ in size/weight, complexity and battery life. The BD FACSCanto™ II flow cytometer is a benchtop analyzer with three different parts (the computer, the fluidic cart and the cytometer). It requires a well-trained technician (3–5 days) with certain skills for preventive maintenance and for acquiring samples, much more space than

2A

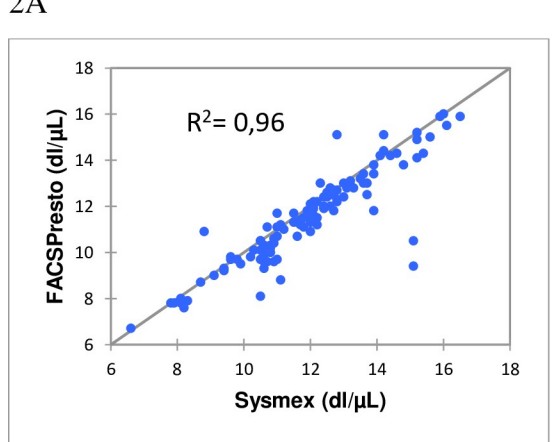

2B

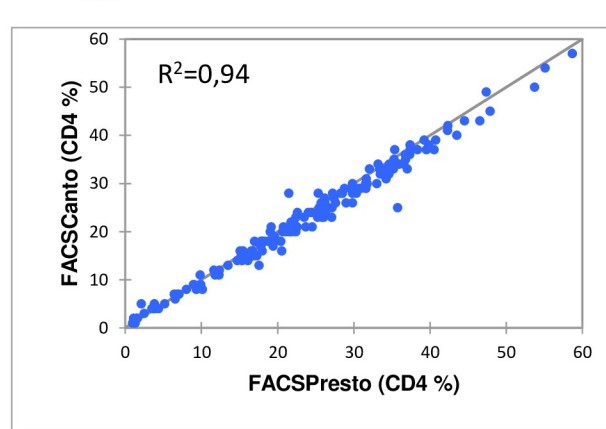

2C

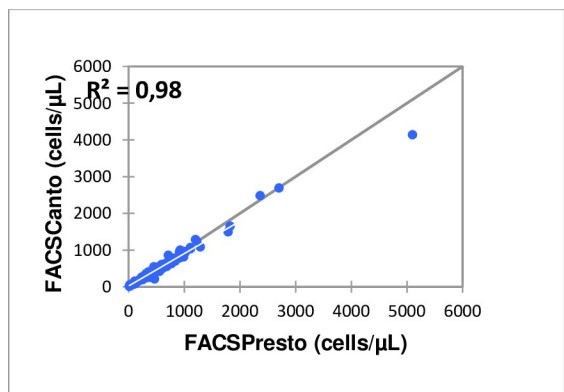

**Fig 2. Deming regression plots for CD4 cell counts, %CD4 and Hb in venous blood.** BDFACSPresto™ vs BD FACSCanto™ II or Sysmex systems. Deming regression results are depicted for venous blood (2A-2C). (2A) The unweighted Deming regression for Hb. (2B) and (2C) The CD4 count and %CD4 cells results respectively, are shown from weighted Deming regression. The y-axis displays the predicate method for CD4 cell counts, or %CD4 cells, and the x-axis corresponds to the BD FACSPresto™ system.

Presto, the use of stable light, at least four different solutions, reagents and consumables, and many steps in the procedure of sample staining. The pre-analytical phase has different steps in comparison to the FACSPresto™. The ease of use, the small size/lightness of weight of the BD FACSPresto™ (less than 7kg) as well as the time to process a sample (approximately 25 minutes, only 4 of which take place in the device), can contribute significantly to improving the turnaround time (TAT) in results to patients. TAT is influenced by the less complex software used with the BD FACSPresto™ combined with its deployability closer to patients being tested. This reduces the need for specimens or patients to be transported to a centralized testing facility for accurate and reliable test results (both diagnostic and monitoring of known infections).

Moreover, another important advantage of FACSPresto™ is the improvement of safety for the operator. The use of capped-cartridge technology on FACSPresto™ minimizes the exposure to open blood tubes containing infectious material. The machine does not require daily maintenance or liquid waste management. BDFACSPresto™ has delivered, in this study, highly accurate results for the routine analysis of lymphocyte CD4 T cells when compared with the widely used conventional flow cytometer BD FACSCanto™ II.

3A

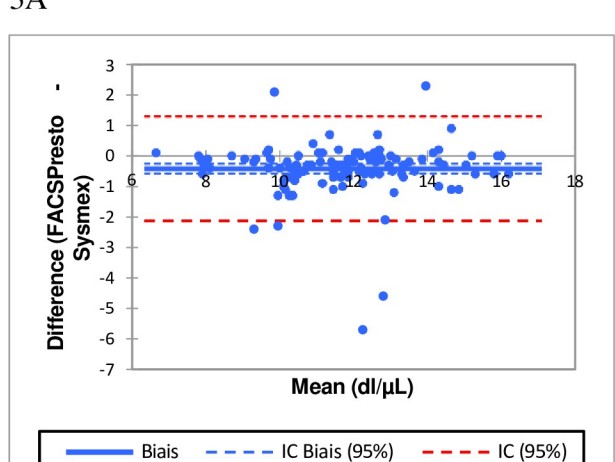

3B

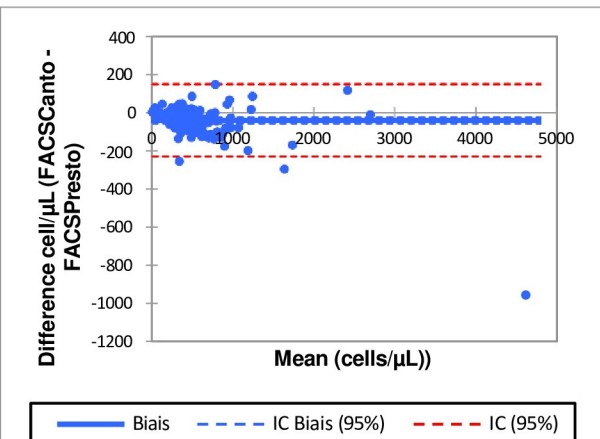

3C

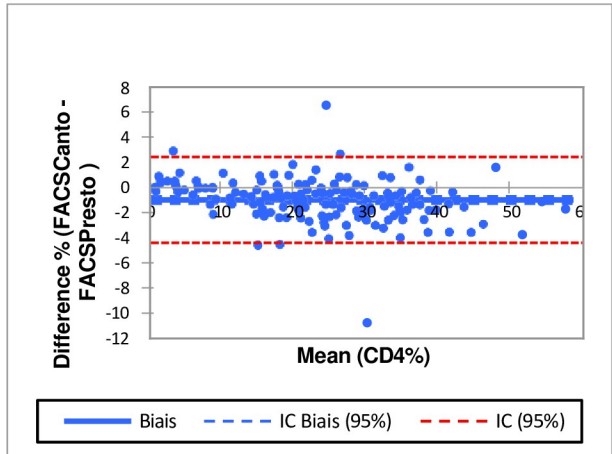

**Fig 3. Bland-Altman plots illustrate the biases for venous blood with limits of agreement for Hemoglobin, CD4 absolute counts, and %CD4.** Biases for hemoglobin concentration in (3A), for % CD4 cell counts are shown in (3B), for CD4 absolute count in (3C). The x-axis displays the average (Hb, CD4 counts, or %CD4 cells) and the y-axis is the difference (Hb, CD4 counts, or %CD4 cells) the solid line represents the mean bias, the dashed line represents mean bias ±1.96SD.

In conclusion, our comparison demonstrated equivalent performance between the BD FACSPresto™ system and the standard-of-care, the BD FACSCanto™ II system, and the Sysmex 100 XN analyzer respectively for enumeration of the CD4 T cells and Hb in human blood anti-coagulated with EDTA from HIV-infected patients.

## Supporting information

**S1 Data.**
(XLSX)

## Acknowledgments

We would like to thank all the study participants. Dr Adrienne MEYER for the review of the manuscript, to Prof Maria Salvato for English grammar and to Irenée DONKAM for the statistical analysis.

We would like to pay tribute remembering the late Professors of Molecular Biology Giancarlo Falcioni and Giancarlo Caulini of the University of Camerino, Italy, as well as Dr. Martin Sobze of the University of Dschang, Cameroon.

## Author Contributions

**Conceptualization:** Bertrand Sagnia, Vittorio Colizzi.

**Data curation:** Bertrand Sagnia, Jules Tchadji.

**Formal analysis:** Ana Gutierez.

**Funding acquisition:** Alexis Ndjolo.

**Investigation:** Sylvie Moudourou, Ana Gutierez.

**Methodology:** Bertrand Sagnia, Fabrice Mbakop Ghomsi, Sylvie Moudourou.

**Project administration:** Vittorio Colizzi.

**Resources:** Sylvie Moudourou, Jules Tchadji, Samuel Martin Sosso.

**Software:** Fabrice Mbakop Ghomsi, Ana Gutierez.

**Supervision:** Samuel Martin Sosso, Alexis Ndjolo.

**Validation:** Bertrand Sagnia, Fabrice Mbakop Ghomsi, Alexis Ndjolo.

**Visualization:** Sylvie Moudourou, Jules Tchadji, Alexis Ndjolo.

**Writing – original draft:** Bertrand Sagnia.

**Writing – review & editing:** Samuel Martin Sosso, Vittorio Colizzi.

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
