## [Decision Letter · Decision Letter 0]

7 Sep 2023

PONE-D-22-35455RACCURATE AND REPRODUCIBLE ENUMERATION OF CD4 T CELL COUNTS AND HEMOGLOBIN LEVELS USING A POINT OF CARE SYSTEM: COMPARISON WITH CONVENTIONAL LABORATORY BASED TESTING SYSTEMS IN A CLINICAL REFERENCE LABORATORY IN CAMEROONPLOS ONE

Dear Dr. Sagnia,

Thank you for submitting your manuscript to PLOS ONE. After careful consideration, we feel that it has merit but does not fully meet PLOS ONE’s publication criteria as it currently stands. Therefore, we invite you to submit a revised version of the manuscript that addresses the points raised during the review process.

We look forward to receiving your revised manuscript.

Kind regards,

Lara Vojnov

Academic Editor

PLOS ONE

Journal Requirements:

 Whilst you may use any professional scientific editing service of your choice, PLOS has partnered with both American Journal Experts (AJE) and Editage to provide discounted services to PLOS authors. Both organizations have experience helping authors meet PLOS guidelines and can provide language editing, translation, manuscript formatting, and figure formatting to ensure your manuscript meets our submission guidelines. To take advantage of our partnership with AJE, visit the AJE website (http://aje.com/go/plos) for a 15% discount off AJE services. To take advantage of our partnership with Editage, visit the Editage website (www.editage.com) and enter referral code PLOSEDIT for a 15% discount off Editage services. If the PLOS editorial team finds any language issues in text that either AJE or Editage has edited, the service provider will re-edit the text for free.

4. Please ensure that you refer to Figure 1 in your text as, if accepted, production will need this reference to link the reader to the figure.

Additional Editor Comments:

In addition to the reviewers' comments, please ensure some explanation and discussion in the rebuttal and within the manuscript on how this present work is different than or builds upon your recent publication:

Sagnia B, Mbakop Ghomsi F, Gutierrez A, Sosso S, Kamgaing R, Nanfack AJ, Nji N, Ambada G, Lissom A, Tchouangueu TF, Ngu Ndengkoh L, Domkam I, Nchinda G, Ndjolo A. Performance of the BD FACSPresto near to patient analyzer in comparison with representative conventional CD4 instruments in Cameroon. AIDS Res Ther. 2020 Aug 17;17(1):53. doi: 10.1186/s12981-020-00309-9. PMID: 32799909; PMCID: PMC7429678.

Reviewers' comments:

Reviewer's Responses to Questions

**Comments to the Author**

1. Is the manuscript technically sound, and do the data support the conclusions?

Reviewer #1: No

Reviewer #2: Partly

2. Has the statistical analysis been performed appropriately and rigorously? 

Reviewer #1: Yes

Reviewer #2: I Don't Know

3. Have the authors made all data underlying the findings in their manuscript fully available?

Reviewer #1: No

Reviewer #2: No

4. Is the manuscript presented in an intelligible fashion and written in standard English?

Reviewer #1: No

Reviewer #2: Yes

5. Review Comments to the Author

Reviewer #1: I have attached my review comments, which are as follows:

Review notes for manuscript entitled: ACCURATE AND REPRODUCIBLE ENUMERATION OF CD4 T CELL COUNTS AND HEMOGLOBIN LEVELS USING A POINT OF CARE SYSTEM: COMPARISONWITH CONVENTIONAL LABORATORY BASED TESTING SYSTEMS IN A

CLINICAL REFERENCE LABORATORY IN CAMEROON

Thank you for presenting this methods comparison paper for review. Broadly, may I start by pointing out that the manuscript that has significant gaps that need to be addressed. There is no data on advantages or usability and yet conclusions have been drawn on usability. There is no data on interoperator variability even though this data was collected in the workflow. There are language barrier problems as well. The results section is incomplete.

The title in the abstract page has a mistake. “Raccurate” should probably be “Accurate”.

Abstract page 7th row entitled “Abstract”, Background section, delete “information about”.

In the whole document, TM should be in superscript.

Although the background suggests that the checked the advantages of using the POC, the manuscript simply compares the performance of the FACSPresto with FACSCanto where CD4 measurements are taken, and with SYSMEX where Hb measurements are taken. Advantages such as cost, time to result, usability, breakdowns etc are not evaluated.

In the results section of the abstract, in the last sentence. I am not convinced that you demonstrated ease of use. For instance, how many steps or preanalytical procedures and analytic procedures were needed to conduct a test?

In the Ethics Statement please replace “routine services CD4 testing” with “routine CD4 testing services”.

Face Page of the manuscript: I feel as though “Vergata” should be followed by a comma and a space before “Rome”.

Overall, the manuscript can benefit from a text editing attempt, to check grammar and syntax.

In the introduction section, second paragraph. “Assess the” … delete the article “the”.

…blood. [5]. Delete the first full stop.

Third paragraph introduction section” “…characteristics like point…” Replace “like’ with “such as”.

This manuscript can benefit from line numbering and page numbering.

Paragraph 4: “…parameters of research and…” should become “…parameters for research and…”

The last sentence in paragraph 4 should be rewritten into two sentences.

In the ‘Materials and Methods’ section, the second sentence is long and unclear and should be revised.

In the samples section, the phrase “…or not processed as per protocol” has an unknown meaning. What or how do you mean when you say that?

‘Analysis on FACSCanto II” is a new subsection and not a continuation of the previous subsection.

Change the case of BD FACSCANTO to BD FACSCanto in the same section.

“…external quality assurance program with the Cameroon…” becomes “… an external quality assurance program with the Cameroon…”

In the Ethical Review section, how did you actually take the informed consent for remnant samples? Had they already been deidentified? Did you go back to the clients?

In the Results section:

“For this study for method comparison…” should be “For this methods comparison study…”

In the Deming Regression paragraph, R2 should be R2

Actually I have an opinion about the results section. I am familiar with describing results carefully and identifying the main findings in a paragraph, after which a figure or a table is introduced. This allows a reader to go through a results section with significant understanding before going to a table or figure. The results section of this manuscript hardly describes the data. Is there a reason for that?

The last sentence in the results section is confusing. What do you mean by “SD of the relative deviation?”.

The first sentence of the Discussions section should be rewritten. There have been some claims in this manuscript about BD FACSCanto having “validated clinical software”. Could you introduce a reference for that?

In the sentence that has reference 17, “…CD4 enumeration still…” should be “…CD4 enumeration is still…”. As a matter of fact, the whole of that sentence should be revised. The sentence that starts with “In addition…” should be rewritten as it is incomplete.

“CD4 results contribute to the interpretation of genotyping…” Why do you think so, or how do you know that? Do you have references you could use to support that?

“The BD FACSPresto results were reproduced when operated by trained operators (but not laboratory technical experts)”. Now, this statement has far-reaching ramifications. Did you speak out of turn? I should like to believe that lab experts are even more likely to find it easy to use.

For the sentence that starts with “The analyser provides absolute…” Please revise the whole sentence to read as follows: “The analyser provides absolute and percentage CD4+ T Cell countsnas well as Hb levels [15] which are all useful for monitoring the paediatric population and tracking anaemia in pregnancy. This instrument has a battery and can run without mains power for about six hours.”

The sentence after that is problematic. It says that The POC device is easy to use also for personnel not working in the lab. Where would those personnel be working? In your study, you did not evaluate that particular capability, and I would be happy to hear why you think so.

“…the loss of followup of patients..” should be “…loss to follow up…”

“…cross training lab staff and representing…” should become “…cross training lab staff and represents…”

“…facing the accreditation process…” Replace “facing” with “undergoing”.

In tha Third last paragraph in Discussions. You suggest in the second last sentence that the BD FACSPresto is simpler to use than BD FACSCanto. Clarify that.

The last paragraph needs to be rewritten, and the last 6 words should be deleted.

The figures are very confusing. Figure 2 is presented before Figure 1. And, Figure two has graphs 1A, 1B and 1C. How is that? And then Figure 3 has graphs 2A, 2b and 2C.

In Figure 1, the last process in the workflow is “Variability between Instrument/Operator, between lot..” These data are not presented in the paper at all.

In Table 1, there are several coordinates in brackets. For example the last line has 20(32, 79). What are those numbers? This issue can also be see in Table 2.

Reviewer #2: Thank you for the opportunity to review the manuscript. CD4 is still a very useful tool for patient management. And as the authors pointed out, access to reliable and accurate CD4 testing in remote settings and lower levels of healthcare can be challenging, yet healthcare providers heavily rely on this tool.

General Comments:

The authors state they compared a CD4 ‘POC’ instrument against conventional lab based technologies. They have successfully shown performance correlations, but perhaps some of the other statements made regarding this POC technology may require more evidence or explanation. See examples below:

• The term ‘POC’ is used somewhat as a catch phrase but probably requires some explanation to the reader. For example, consider that there is now at least one device free technology for CD4 measurement, POC devices that have been successfully used at primary levels of the healthcare system, and near POC instruments best placed at district levels (e.g. require more training and oversight). Could the author validate their use of the term for ‘POC’ with regards to the FACSPresto.

• I wonder if the benefits and limitations of a device free CD4 POC test versus a near POC instrument merits some discussion. As the authors have previously published on other CD4 ‘POC’ devices, perhaps in the discussion an example of where the technologies evaluated here may fit within the Cameroon healthcare system, and where gaps remain.

• Parts of the discussion belong in the background or methods.

• 11 samples were excluded due to instrument problems or protocol not followed. Could the authors expand on this? This could impact the results and findings.

• Results are a bit thin - e.g. Table 1 needs to be described.

• pg 14: Authors state ‘The BD FACSPresto results were reproducible when operated by trained operators (but not laboratory technical experts) such as nurses after minimum training (personal data).’ Then follow with a conflicting statement: ‘easy to use attesting the fact that BD FACSPresto can accurately identify those patients with ART failure irrespective of technical laboratory expertise.’ Can you please clarify?

Minor comments:

• pg 9: ‘many pharmaceutical companies proposed new instruments. Should this be diagnostic companies?

• Is there EQA for the FACSPresto? EQA was mentioned for the other instruments.

• Numbered figures do not match the text box.

6. PLOS authors have the option to publish the peer review history of their article (what does this mean?). If published, this will include your full peer review and any attached files.

Reviewer #1: **Yes: **Matilu Mwau

Reviewer #2: No

---

## [Author Response · Author response to Decision Letter 0]

27 Sep 2023

Review notes for manuscript entitled: ACCURATE AND REPRODUCIBLE ENUMERATION OF CD4 T CELL COUNTS AND HEMOGLOBIN LEVELS USING A POINT OF CARE SYSTEM: COMPARISONWITH CONVENTIONAL LABORATORY BASED TESTING SYSTEMS IN A

CLINICAL REFERENCE LABORATORY IN CAMEROON

Thank you for presenting this methods comparison paper for review. Broadly, may I start by pointing out that the manuscript that has significant gaps that need to be addressed. There is no data on advantages or usability and yet conclusions have been drawn on usability. There is no data on interoperator variability even though this data was collected in the workflow. There are language barrier problems as well. The results section is incomplete.

Response: Thank you for your remarks/ Prof Maria SALVATO has improve the English language of the manuscript

Maria S. Salvato, PhD

expert in English grammar

Adj. Professor, Department of Veterinary Medicine

8075 Greenmead Drive,

University of Maryland

College Park, MD 20742

410-493-8912

email: msalvato8@gmail.com; msalvat1@umd.edu

The title in the abstract page has a mistake. “Raccurate” should probably be “Accurate”.

Abstract page 7th row entitled “Abstract”, Background section, delete “information about”. 

Response: Modified in the manuscript

In the whole document, TM should be in superscript.

Although the background suggests that the checked the advantages of using the POC, the manuscript simply compares the performance of the FACSPresto with FACSCanto where CD4 measurements are taken, and with SYSMEX where Hb measurements are taken. Advantages such as cost, time to result, usability, breakdowns etc are not evaluated.

Response: Thank you for your observation, we inserted it the manuscript

In the results section of the abstract, in the last sentence. I am not convinced that you demonstrated ease of use. For instance, how many steps or preanalytical procedures and analytic procedures were needed to conduct a test?

Response: Comparing both instrument, steps in the pre analytical procedure is reduced comparing to BD FACSCanto II, because you need to do the fluidic start-up, after you have to calibrate the instrument with the 7 colors set-up for clinical software. In the FACSPresto, we have to carry one drop of sample in the cartridges and capp. When you turn on the instrument, we do not need beads, you have an internal quality control done by the instrument and if it’s in the range, the instrument is ready to read samples

In the Ethics Statement please replace “routine services CD4 testing” with “routine CD4 testing services”.

Response: Modified in the manuscript

Face Page of the manuscript: I feel as though “Vergata” should be followed by a comma and a space before “Rome”.

Response: Modified in the manuscript

Overall, the manuscript can benefit from a text editing attempt, to check grammar and syntax.

In the introduction section, second paragraph. “Assess the” … delete the article “the”.

…blood. [5]. Delete the first full stop.

Response: Modified in the manuscript

Third paragraph introduction section” “…characteristics like point…” Replace “like’ with “such as”.

Response: Modified in the manuscript

This manuscript can benefit from line numbering and page numbering.

Paragraph 4: “…parameters of research and…” should become “…parameters for research and…”

Response: Modified in the manuscript

The last sentence in paragraph 4 should be rewritten into two sentences.

Modified in the manuscript

In the ‘Materials and Methods’ section, the second sentence is long and unclear and should be revised. 

Modified in the manuscript

In the samples section, the phrase “…or not processed as per protocol” has an unknown meaning. What or how do you mean when you say that?

Response: Thank you for your observation. This means that, if the sample is clotted or the volume is not sufficient, it will not be processed 

‘Analysis on FACSCanto II” is a new subsection and not a continuation of the previous subsection.

Response: You are wright 

Change the case of BD FACSCANTO to BD FACSCanto in the same section.

Modified in the manuscript

“…external quality assurance program with the Cameroon…” becomes “… an external quality assurance program with the Cameroon…”

Modified in the manuscript

In the Ethical Review section, how did you actually take the informed consent for remnant samples? Had they already been deidentified? Did you go back to the clients?

Response: Thank you for your observation: We informed the participants who came for their routine examinations that the remnant of their sample, if they consent, will be used to carry out a comparative study between two devices. Samples received at the laboratory receive a code. It is decoded only when printing and rendering the results

In the Results section:

“For this study for method comparison…” should be “For this methods comparison study…”

Modified in the manuscript

In the Deming Regression paragraph, R2 should be R2

Modified in the manuscript

Actually I have an opinion about the results section. I am familiar with describing results carefully and identifying the main findings in a paragraph, after which a figure or a table is introduced. This allows a reader to go through a results section with significant understanding before going to a table or figure. The results section of this manuscript hardly describes the data. Is there a reason for that?

Response: Thank you for the observation. No there is no reason for that. 

The last sentence in the results section is confusing. What do you mean by “SD of the relative deviation?”.

Response: Modified in the manuscript as standard deviation

The first sentence of the Discussions section should be rewritten. There have been some claims in this manuscript about BD FACSCanto having “validated clinical software”. Could you introduce a reference for that?

Response: I have cancelled it in the manuscript! I was confused with the validated method of CD4 enumeration multitest with another not the validated software

In the sentence that has reference 17, “…CD4 enumeration still…” should be “…CD4 enumeration is still…”. As a matter of fact, the whole of that sentence should be revised. The sentence that starts with “In addition…” should be rewritten as it is incomplete. 

Response: Modified in the manuscript

“CD4 results contribute to the interpretation of genotyping…” Why do you think so, or how do you know that? Do you have references you could use to support that?

Response: In the International research centre where I’m working, we have a Virology Lab which is doing gentoping test for patients. Our country is following the WHO guidelines. In these guidelines, CD4 count testing are not necessary and there is no funds for that. CD4 count are not done in all the country except my centre and a confessional health facility. It’s demonstrate that Routinely, the CD4 level can often indicate the urgency of putting the patient on a treatment with strong antiviral activity (sometimes quadruple therapy) in order to quickly reduce viremia and restore the CD4 level.

“The BD FACSPresto results were reproduced when operated by trained operators (but not laboratory technical experts)”. Now, this statement has far-reaching ramifications. Did you speak out of turn? I should like to believe that lab experts are even more likely to find it easy to use. 

Response: The non-lab personnel is still using it with a less than one day training

For the sentence that starts with “The analyser provides absolute…” Please revise the whole sentence to read as follows: “The analyser provides absolute and percentage CD4+ T Cell countsnas well as Hb levels [15] which are all useful for monitoring the paediatric population and tracking anaemia in pregnancy. This instrument has a battery and can run without mains power for about six hours.”

Response: Modified in the manuscript

The sentence after that is problematic. It says that The POC device is easy to use also for personnel not working in the lab. Where would those personnel be working? In your study, you did not evaluate that particular capability, and I would be happy to hear why you think so.

Response: In our facility, we have a Medical section unit where we receive patient. During the one day training for this evaluation, the personnel not working in the lab, was associated and they analysed some samples, repeated by the lab personnel: the correlation was more than 99%, 

“…the loss of followup of patients..” should be “…loss to follow up…”

Response: Modified in the manuscript

“…cross training lab staff and representing…” should become “…cross training lab staff and represents…”

Response: Modified in the manuscript

“…facing the accreditation process…” Replace “facing” with “undergoing”.

Response: Modified in the manuscript

In tha Third last paragraph in Discussions. You suggest in the second last sentence that the BD FACSPresto is simpler to use than BD FACSCanto. Clarify that.

Response: Thank you for your observation. I inserted in the manuscript few sentences about that saying that to use the BD FACSCanto II, it’s necessary to have some skills, to have a certain level of education. As I said, we have the software (FACSCanto Clinical software) many procedures (start up, set up, shutdown, cleaning instrument, samples staining), two reagents in the samples staining (antibodies, FACSLysing solution), many solutions (FACSFlow, Rinse, Clean, Shutdown) make noise) etc. It’s also expensive 

The last paragraph needs to be rewritten, and the last 6 words should be deleted.

Response: Modified in the manuscript

The figures are very confusing. Figure 2 is presented before Figure 1. And, Figure two has graphs 1A, 1B and 1C. How is that? And then Figure 3 has graphs 2A, 2b and 2C.

Response: Modified in the manuscript

In Figure 1, the last process in the workflow is “Variability between Instrument/Operator, between lot..” These data are not presented in the paper at all.

Response: Thank you for your observation. Because of the fact that there was not significant differences, this is why we estimate not necessary to talk about

In Table 1, there are several coordinates in brackets. For example the last line has 20(32, 79). What are those numbers? This issue can also be see in Table 2.

Response: Thank you very much for your remark too important for the comprehension. Some brackets are representing the percentage, others represent the range. It’s modified in the manuscript

---

## [Decision Letter · Decision Letter 1]

15 Dec 2023

PONE-D-22-35455R1RACCURATE AND REPRODUCIBLE ENUMERATION OF CD4 T CELL COUNTS AND HEMOGLOBIN LEVELS USING A POINT OF CARE SYSTEM: COMPARISON WITH CONVENTIONAL LABORATORY BASED TESTING SYSTEMS IN A CLINICAL REFERENCE LABORATORY IN CAMEROONPLOS ONE

Dear Dr. Sagnia,

Thank you for submitting your manuscript to PLOS ONE. After careful consideration, we feel that it has merit but does not fully meet PLOS ONE’s publication criteria as it currently stands. Therefore, we invite you to submit a revised version of the manuscript that addresses the points raised during the review process. **Minor revision is requested.** Not all comments of Reviewer 2 have been addressed. Below are additional comments: 

We look forward to receiving your revised manuscript.

Kind regards,

Seth Agyei Domfeh

Academic Editor

PLOS ONE

Journal Requirements:

Reviewers' comments:

Reviewer's Responses to Questions

**Comments to the Author**

1. If the authors have adequately addressed your comments raised in a previous round of review and you feel that this manuscript is now acceptable for publication, you may indicate that here to bypass the “Comments to the Author” section, enter your conflict of interest statement in the “Confidential to Editor” section, and submit your "Accept" recommendation.

Reviewer #1: All comments have been addressed

Reviewer #2: No

2. Is the manuscript technically sound, and do the data support the conclusions?

Reviewer #1: Yes

Reviewer #2: Partly

3. Has the statistical analysis been performed appropriately and rigorously? 

Reviewer #1: Yes

Reviewer #2: N/A

4. Have the authors made all data underlying the findings in their manuscript fully available?

Reviewer #1: Yes

Reviewer #2: Yes

5. Is the manuscript presented in an intelligible fashion and written in standard English?

Reviewer #1: Yes

Reviewer #2: Yes

6. Review Comments to the Author

Reviewer #1: My suggestions have now been addressed and the manuscript is much better. I have no further comments to make.

Reviewer #2: I have reviewed the revised manuscript by Sagnia et al on the evaluation of a CD4 POC for CD4 T cell counts and hemoglobin levels in Cameroon. Unfortunately, the manuscript still requires major revisions. Despite the potentially quality work done, there were some major gaps in the data and the overall messaging was unclear.

From what I understood, the authors were comparing the Canto to the Caliber, which measures CD4 and hemoglobin, but is lab based, and the Presto, which also measures both, as a POC. Even this was somewhat confusing as they briefly introduced the assays. Given the overlap in assays, perhaps a stronger statement on the ‘why’ the comparison and clearer descriptions of the technologies, would have given the reader a better understanding on what I assume to be the purpose of such an evaluation – the need for POC instruments that can report both CD4 counts and hemoglobin levels for use at ANC settings. And that an additional quality product provides a more competitive market and healthier landscape.

There were gaps in the Results section.

- Table 1 was introduced but not discussed. Were there any comments or statements regarding the characteristics of study participants?

- “11 samples were excluded from analysis because of instrument problems or not processed as per protocol described above.” This seems important given the authors are evaluating the instrument. What exactly were the problems with these 11 samples? What were the instrument problems? And why were the samples not processed per protocol? Are these a potential issues with the use and roll out of this instrument?

- The authors state “During the analysis, we observed that for CD4 less than 350 cells/mL, the FACSPrestoTM tends to underestimate the values compared to the FACSCantoTM II.” This is an important observation and should be further discussed. Is this a true and significant observation? What then are the implications of using Presto vs Canto?

- Overall, the results section was thin and could use with more explained observations that begins to let us know how these observations could impact the Cameroon CD4 program.

The Discussion section was also thin on meaningful commentary.

- As mentioned above, there were quite a few observations that could be further discussed.

- Another example is the opening statement on the “first study done in Cameroon on the BD FACSPrestoTM performance was published comparing BD FACSPrestoTM versus PIMA, BD FACSCountTM and BD FACSCaliburTM.” What were the findings from this study? Why did the authors bring this up in the Discussion?

- Much of what has been included in this section belongs in the Intro or Methods sections.

- Beginning with Line 2019, the authors make what is probably the ‘take home’ message: “The present study is the first published paper to demonstrate that BD FACSPrestoTM provides excellent results when compared with a BD FACSCantoTM II clinical machine, showing very good agreement between both instruments.” But because the results were so thin, it leaves the reader questioning ‘where is the data to support this?’.

- Overall, the Discussions section needs to be rewritten. Perhaps each of the statements made by the authors could immediately be followed by a summary of the results used to draw this conclusion and then their discussion on this, with each having its own paragraph.

- Probably too much detail was put in this section describing product characteristics, but not enough to conclude “our comparison demonstrated equivalent performance between the BD FACSPrestoTM system and the standard-of-care, the BD FACSCantoTM II system, and the Sysmex 100 XN analyzer…”.

Minor comments

- Line 222 – “This study supports that CD4 T cells measurements were not influenced by the counting method used.” What does this mean? What counting method?

- Line 233 – “Placing this instrument in rural areas will reduce the loss to follow-up…” Test time to result or turn-around time has not yet been mentioned.

- Line 241 – “Based on internal experience,…” This is not internal experience, but a product characteristic.

- Line 253 – “This reduces the need for specimens or patients to be transported to a centralized testing facility for accurate and reliable test results (both diagnostic and monitoring of known infections).” Is CD4 being used to monitor infections (surveillance?)? Or perhaps monitoring patient treatment?

7. PLOS authors have the option to publish the peer review history of their article (what does this mean?). If published, this will include your full peer review and any attached files.

Reviewer #1: **Yes: **Prof. Matilu Mwau

Reviewer #2: No

---

## [Author Response · Author response to Decision Letter 1]

2 Jan 2024

6. Review Comments to the Author

Reviewer #1: My suggestions have now been addressed and the manuscript is much better. I have no further comments to make.

Reviewer #2: I have reviewed the revised manuscript by Sagnia et al on the evaluation of a CD4 POC for CD4 T cell counts and hemoglobin levels in Cameroon. Unfortunately, the manuscript still requires major revisions. Despite the potentially quality work done, there were some major gaps in the data and the overall messaging was unclear.

From what I understood, the authors were comparing the Canto to the Caliber, which measures CD4 and hemoglobin, but is lab based, and the Presto, which also measures both, as a POC. Even this was somewhat confusing as they briefly introduced the assays. Given the overlap in assays, perhaps a stronger statement on the ‘why’ the comparison and clearer descriptions of the technologies, would have given the reader a better understanding on what I assume to be the purpose of such an evaluation – the need for POC instruments that can report both CD4 counts and hemoglobin levels for use at ANC settings. And that an additional quality product provides a more competitive market and healthier landscape.

There were gaps in the Results section.

- Table 1 was introduced but not discussed. Were there any comments or statements regarding the characteristics of study participants?

Response: We modified in the manuscript

- “11 samples were excluded from analysis because of instrument problems or not processed as per protocol described above.” This seems important given the authors are evaluating the instrument. What exactly were the problems with these 11 samples? What were the instrument problems? And why were the samples not processed per protocol? Are these a potential issues with the use and roll out of this instrument?

Response: 11 samples were excluded because after de test on the FACSPresto, there were no results but the code 6102 was release, which recommended to rerun the quality control and to retest de sample. This was done and after all, no results. On the FACSCanto II, for these 11 samples, the CD4 were less than 30 cells/µL. These values are out of the validated range on the FACSPresto.

- The authors state “During the analysis, we observed that for CD4 less than 350 cells/mL, the FACSPrestoTM tends to underestimate the values compared to the FACSCantoTM II.” This is an important observation and should be further discussed. Is this a true and significant observation? What then are the implications of using Presto vs Canto?

Response: We modified in the manuscript

. 

- Overall, the results section was thin and could use with more explained observations that begins to let us know how these observations could impact the Cameroon CD4 program.

Response: We modified in the manuscript

The Discussion section was also thin on meaningful commentary.

- As mentioned above, there were quite a few observations that could be further discussed.

- Another example is the opening statement on the “first study done in Cameroon on the BD FACSPrestoTM performance was published comparing BD FACSPrestoTM versus PIMA, BD FACSCountTM and BD FACSCaliburTM.” What were the findings from this study? Why did the authors bring this up in the Discussion?

Response: We modified in the manuscript

- Much of what has been included in this section belongs in the Intro or Methods sections.

- Beginning with Line 2019, the authors make what is probably the ‘take home’ message: “The present study is the first published paper to demonstrate that BD FACSPrestoTM provides excellent results when compared with a BD FACSCantoTM II clinical machine, showing very good agreement between both instruments.” But because the results were so thin, it leaves the reader questioning ‘where is the data to support this?’.

Response: We modified in the manuscript

- Overall, the Discussions section needs to be rewritten. Perhaps each of the statements made by the authors could immediately be followed by a summary of the results used to draw this conclusion and then their discussion on this, with each having its own paragraph.

- Probably too much detail was put in this section describing product characteristics, but not enough to conclude “our comparison demonstrated equivalent performance between the BD FACSPrestoTM system and the standard-of-care, the BD FACSCantoTM II system, and the Sysmex 100 XN analyzer…”.

Response: Thank you for your observation. We compared the correlation between the results obtained from these two instruments for CD4 enumeration 

Minor comments

- Line 222 – “This study supports that CD4 T cells measurements were not influenced by the counting method used.” What does this mean? What counting method?

Response: This means that if you test a patient on a FACSPresto or a FACSCanto II, the results will be the same. 

- Line 233 – “Placing this instrument in rural areas will reduce the loss to follow-up…” Test time to result or turn-around time has not yet been mentioned.

You are right, this has to be mentioned in the manuscript

- Line 241 – “Based on internal experience,…” This is not internal experience, but a product characteristic.

You are right, it’s the characteristic of the product

- Line 253 – “This reduces the need for specimens or patients to be transported to a centralized testing facility for accurate and reliable test results (both diagnostic and monitoring of known infections).” Is CD4 being used to monitor infections (surveillance?)? Or perhaps monitoring patient treatment?

Response: In the context of HIV/AIDS, CD4 count is used to monitoring the patient treatment. We modified in the manuscript

7. PLOS authors have the option to publish the peer review history of their article (what does this mean?). If published, this will include your full peer review and any attached files.

Do you want your identity to be public for this peer review? For information about this choice, including consent withdrawal, please see our Privacy Policy.

Reviewer #1: Yes: Prof. Matilu Mwau

Reviewer #2: No

Response: We used it to transform Figures

---

## [Decision Letter · Decision Letter 2]

15 Jan 2024

RACCURATE AND REPRODUCIBLE ENUMERATION OF CD4 T CELL COUNTS AND HEMOGLOBIN LEVELS USING A POINT OF CARE SYSTEM: COMPARISON WITH CONVENTIONAL LABORATORY BASED TESTING SYSTEMS IN A CLINICAL REFERENCE LABORATORY IN CAMEROON

PONE-D-22-35455R2

Dear Dr. Sagnia,

We’re pleased to inform you that your manuscript has been judged scientifically suitable for publication and will be formally accepted for publication once it meets all outstanding technical requirements.

Kind regards,

Seth Agyei Domfeh, PhD

Academic Editor

PLOS ONE

Reviewers' comments:

Reviewer's Responses to Questions

**Comments to the Author**

1. If the authors have adequately addressed your comments raised in a previous round of review and you feel that this manuscript is now acceptable for publication, you may indicate that here to bypass the “Comments to the Author” section, enter your conflict of interest statement in the “Confidential to Editor” section, and submit your "Accept" recommendation.

Reviewer #2: All comments have been addressed

2. Is the manuscript technically sound, and do the data support the conclusions?

Reviewer #2: Yes

3. Has the statistical analysis been performed appropriately and rigorously? 

Reviewer #2: N/A

4. Have the authors made all data underlying the findings in their manuscript fully available?

Reviewer #2: Yes

5. Is the manuscript presented in an intelligible fashion and written in standard English?

Reviewer #2: Yes

6. Review Comments to the Author

Reviewer #2: Comments have been addressed. Thank you again for your time with revising the manuscript. This is an important and timely topic.

---

## [Editor Report · Acceptance letter]

24 Feb 2024

PONE-D-22-35455R2 

PLOS ONE

Dear Dr. Sagnia, 

I'm pleased to inform you that your manuscript has been deemed suitable for publication in PLOS ONE. Congratulations! Your manuscript is now being handed over to our production team.

Kind regards, 

on behalf of

Dr. Seth Agyei Domfeh 

Academic Editor

PLOS ONE